# Man-Made Raw Materials for the Production of Composite Silicate Materials Using Energy-Saving Technology

Sultan Auyesbek [1], Nuraly Sarsenbayev [1], Aisulu Abduova [2], Bakhytzhan Sarsenbayev [1], Saken Uderbayev [3], Zhambyl Aimenov [4], Gulmira Kenzhaliyeva [2], Uzakbai Akishev [5], Taslima Aubakirova [1], Gaukhar Sauganova [1], Eldar Amanov [1], Olga Kolesnikova [6,*] and Igor Panarin [7,*]

1   Research Laboratory of Building Materials, Construction and Architecture, M. AuezovSouth Kazakhstan University, Shymkent 160012, Kazakhstan
2   Department of Ecology, M. Auezov, South Kazakhstan University, Shymkent 160012, Kazakhstan
3   Department of Architecture and Construction Production, Korkyt Ata Kyzylorda University, Kyzylorda 160012, Kazakhstan
4   Scientific Research Institute of Natural and Technical Sciences, M. Auezov, South Kazakhstan University, Shymkent 160012, Kazakhstan
5   Department of Technical Disciplines, Kazakh-Russian International University, Aktobe 160015, Kazakhstan
6   Department of Science, Production and Innovation, M. Auezov South Kazakhstan University, Shymkent 160012, Kazakhstan
7   Polytechnic Institute, Far Eastern Federal University, 690922 Vladivostok, Russia
*   Correspondence: ogkolesnikova@yandex.kz (O.K.); panarin.ii@dvfu.ru (I.P.); Tel.: +7-7052566897 (O.K.)

**Abstract:** This paper presents the development of composite silicate mass compositions based on man-made waste for the production of autoclave hardening products, as well as the results of physico-chemical studies of hydration products of silicate materials. The possibility, expediency and efficiency of using multi-tonnage technogenic waste of Kazakhstan in the industry of composite building materials is shown. Based on the results of the conducted research, the composition of a composite silicate mass based on burnt carbonate-barium tailings (8–12%), electrothermophosphoric slags (82–90%) or sand and dust from cement kiln electrofilters (2–5%) for the production of autoclave hardening products was developed. It was found that the cementing substance in composite silicate materials is represented by CSH(B) calcium silicate hydrates, tobermorite and serpentine. The simultaneous presence of fibrous and crystalline calcium and magnesium silicate hydrates in hydration products leads to the creation of composite products with a maximum strength of 41–49 MPa.

**Keywords:** dolomite-barium tailings; granular electrothermophosphoric slags; dust from electrofilters; silicate bricks; man-made waste; composite materials

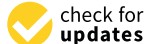



## 1. Introduction

Increasing the efficiency of the use of all types of resources is a strategic goal of any modern society. The final criterion for choosing the right path toward this goal is to increase the comfort of the human environment, including increasing the levels of well-being, health and safety. An important tool for achieving this is the development of the construction complex, which is impossible without the creation of effective high-tech materials and technologies oriented to a wide practice, which is defined in the strategy "Kazakhstan–2050" as one of the priority directions: the transition to the innovative industry.

The materials traditionally used in civil construction are well mastered and studied; however, having been developed 20–30 years ago or more, they have exhausted their potential to increase the efficiency of buildings. Thus, for residential buildings in the northern regions of Kazakhstan (the reduced heat transfer resistance of the outer walls is 3.5 °C/W), the thickness of a single-layer enclosing structure based on expanded clay concrete with an average density of 700–800 kg / m3 should be 0.7–1.4 m, and for silicate gas with an average density of 500 kg/m$^3$, it should be 0.4–0.5 m. The results demonstrated by expanded clay

concrete cannot be considered satisfactory, and the disadvantage of gas-silicate bricks is the expediency of its production only at large enterprises, which significantly increases the cost of its transportation to the construction area, creates prerequisites for monopoly pricing and prevents the introduction of innovations in the production of building materials. In addition, the materials considered do not allow us to obtain comprehensive solutions for other structures, except for load-bearing walls.

The improvement of heat-saving and acoustic properties is now mainly realized through a combination of various materials, which inevitably leads to an increase in the cost and a decrease in the pace of construction, and creates reliability and safety risks. The issues of protecting people from negative external and internal influences in ordinary buildings remain practically unresolved, therefore not meeting the objectives of improving the health of the nation.

A comprehensive solution to the problem is the development, based on comprehensive theoretical research and analysis of existing experience, of a wide range of modern composite materials based on natural and man-made raw materials. Their distinctive features are their focus on the preferential use of local raw materials, flexibility in the organization of production and application and high knowledge intensity and modernization potential.

Technological processes of the mining and processing industry are associated with the formation of production waste and require bringing them to condition. Therefore, the problem of the integrated use of mineral raw materials and industrial waste disposal, with the creation of energy-saving, environmentally friendly technologies focused on the production of several types of products in conditions of an acute shortage of fuel and energy resources, is of decisive importance and is an urgent problem at the present stage.

To solve the problem of waste disposal, an integrated approach is required: theoretical and practical developments and proposals are needed that allow for the use of man-made products of enterprises as valuable raw materials for obtaining marketable products in various industries. The development of new materials and technologies is an objective necessity of the industrial and innovative development of Kazakhstan. The proposed scenario for the development of the "New Materials and Technologies" direction in Kazakhstan until 2030 assumes a progressive and dynamic development of the economy of the Republic of Kazakhstan that allows for gradually commissioning new energy and production facilities, focusing on new resource-saving technologies, taking into account the secured demand for new composite materials and presenting the possibility of transition to a "green" economy and sustainable development of the country [1–7].

The development of composite materials with high-performance properties and new functionalities is an important factor in solving environmental problems, such as the development of natural and man-made raw materials in Kazakhstan and the creation of new resource-saving technologies [1,2,8–10].

Work is being successfully carried out to replace lime with more stable, cheap and efficient raw materials: by-products of the industry and rocks of the required composition.

When creating the theoretical foundations and technology for the production of composite building materials from by-products, the following principles were used in the work:

- Ensuring the maximum material intensity of the developed technological solutions based on the use of secondary raw materials of the region [8–14];
- Obtaining a high technical and economic efficiency of the developed technologies and composite materials;
- Compliance with the requirements of ecology and protection of the biosphere, conservation of natural resources, useful agricultural areas, atmosphere and reduction in costs for the maintenance of slag and tailings dumps.

Silicate brick currently ranks third among small-piece wall materials in terms of demand [11,12]. According to the nearest forecast [13], it will not only retain its market niche in the future, but its production will receive further development at the modern

technological level. Thus, the purpose of these studies was to develop a composite material: silicate brick with the use of technogenic waste as a secondary raw material.

## 2. Materials and Methods

In order to obtain a composite material in the form of silicate bricks, the following technogenic wastes were used as raw materials: dolomite-barium tailings of JSC "Ach-polymetal" of the Khantagitailings dump (Khantagi, Kazakhstan), granular electrother-mophosphoric slags—calcium of silicate waste from phosphorus sublimation during elec-trothermal processing of Karatau phosphorites at the Novo-Dzhambul phosphorus plant (Taraz, Kazakhstan) Taraz—captured dust on electrofilters from the cement plant of JSC "Shymkent Cement" (Shymkent, Kazakhstan) and natural raw materials in the form of sand from the Volsky deposit (Volsk, Russia).

Polymetallic ore dressing wastes—carbonato-barium "tails" —are a finely ground product that does not require additional grinding before use (Figure 1).

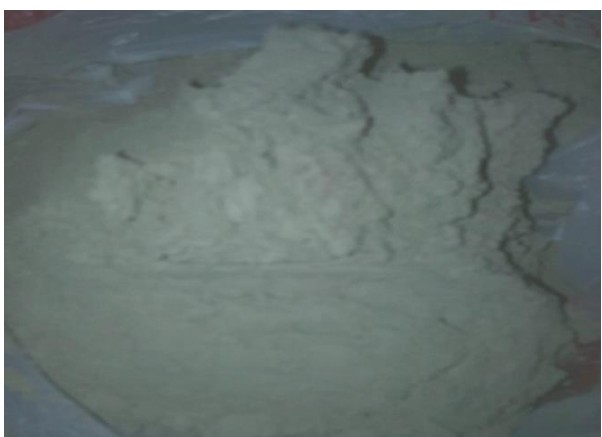

**Figure 1.** Appearance of polymetallic ore dressing waste.

Granulated electrothermophosphoric slag of the Novodzhambul phosphorus plant (Taraz, Kazakhstan) meets the requirements of GOST 3476-74. The content of silica and calcium oxide in phosphoric slag was 41.5–42.5%, fluorine 1.7%.

The Volsky sand used was gray with a yellowish tinge, feldspar-quartz, fine with slightly rounded grains. Sand by chemical composition contained: $SiO_2$—77.9%; $SO_3$—0.35%; $Na_2O + KO$—5.6%; $CaO$—3.4%; $MgO$—1.38%; $Fe_2O_3$—3.02%; $Al_2O_3$—10.6%. According to laboratory studies, the grain composition of sand meets the requirements of GOST 8736-93 and was represented by the following fractions: 1.25 mm—1.3%, 0.63 mm—22.53%, 0.315 mm—64.3%, 0.14 mm—12.02%. The size modulus is $M_d = 0.72–1.25$.

The dust used by the electric filters of the cement plant "Shymkent Cement" was represented by the chemical composition, which is shown in Table 1.

**Table 1.** Chemical composition of dust from electrofilters.

| Chemical Compounds | Content of Oxides, Mass.% |
|---|---|
| $SiO_2$ | 14.59 |
| $Al_2O_3$ | 3.75 |
| $Fe_2O_3$ | 2.71 |
| $CaO$ | 42.05 |
| $MgO$ | 0.60 |
| $K_2O$ | 7.39 |
| $Na_2O$ | 0.97 |
| $SO_3$ | 0.30 |
| p.p.p. | 27.64 |

Calcium lime with a magnesium oxide content of no more than 5% is used for the production of autoclave building materials and products. According to geological surveys, approximately 60% of limestones do not meet these requirements and are highly magnesian. The demand for pure limestones is increasing, and the reserves of proven calcium limestones are decreasing. Therefore, the use of lime containing more than 5% magnesium oxide is of great economic importance. It should be noted that high-magnesian limestones and dolomites are located more evenly than pure calcium.

At the present stage, autoclave technology is progressively developing. It has large reserves for increasing the volume of production, the productivity of workers and the quality of products, improving their aesthetic properties and reducing the cost. Work is successfully being carried out to replace lime with more stable, cheap and efficient raw materials: by-products of industry and rocks of proper composition.

The problems of obtaining composite silicate material in the form of silicate bricks from the waste of a number of industries were solved with the involvement of a complex of physico-chemical and physico-mechanical analysis methods. Thus, when performing studies to determine the specific surface area, the determination was carried out by the method of breathability on the PSX-12 device (St. Petersburg, Russia). X-ray diffraction analysis is the most versatile and perfect method of materials research in comparison with other physico-chemical methods of analysis. The main advantage of this method is the study of a solid in an unchanged state: as a result of the analysis, the substance or its components are determined directly, and, if necessary, individual modifications can be identified (for example, $CaCO_3$—calcite or aragonite). The method has a high sensitivity to individual minerals in order to significantly reduce the analysis time and the ability to determine the presence of a particular mineral if its content in the material is less than 3%. To obtain an X-ray image, a sample in an amount of approximately 1–2 g was ground in a mortar until it completely passed through a sieve. The X-ray images were taken on a DRON-3 device (Moscow, Russia) with a copper anti-cathode and a nickel filter in the range of double reflection angles of 2θ5-64 degrees on filtered SIKA radiation with a Ni filter, at a tube voltage of 20 kV, where the anode current of the tube is 20 mA, the measurement limit is from 1000 to 4000 imp/s and the speed of the counter is 4 degrees/min, with an angular mark of $-1°$.

Scanning electron microscopy analysis was carried out on an optical scientific microscope NU-2E by Carl Zeiss (Jena, Germany) with magnification from 5 to 2500 times.

Physico-mechanical studies of composite material in the form of silicate bricks were studied using a hydraulic press of the PGM-100-MG4-A brand developed by Stroypribor (St. Petersburg, Russia).

Laboratory studies were carried out using modern equipment on the basis of the Department of Cement, Ceramics and Glass Technology (M. Auezov SKU, Shymkent. RK), (V. G. Shukhov BSTU, Belgorod, Russia), the Testing Regional Laboratory of Engineering profile "Structural and Biochemical Materials" and the Research Institute "Building Materials, Construction and Architecture" (M. Auezov SKU, Shymkent, Kazakhstan).

### 3. Results and Discussion

Currently, the use of dolomitized lime in the silicate industry is not a new issue. Klimovichesky and Orsha plants use dolomite lime, producing silicate materials that meet the requirements of the standard. E.D. Pevsner recommends firing dolomitized limestone at 850–900 °C for one to three hours with rapid cooling [14–22].

The firing of the tailings at 750–1300 °C with an interval of 50–100 °C and an X-ray phase analysis of the firing products showed that the optimum temperature should be considered as 800–900 °C. The fired material has the largest specific surface area, and the lime concentration reaches maximum values with a relatively weakly expressed recrystallization of magnesium oxide. The content of the latter in the burnt waste is 22%: dust from the electrofilters of the cement plant.

Due to the intensification of the clinker firing process in rotary kilns and the commissioning of electric filters in cement plants, the amount of dust captured has significantly increased. As a rule, the carbonate component is carried away to a large extent, the clay component to a lesser extent and iron-containing additives, clinker phases (10–15%) and alkalis (potassium and sodium sulfates) are carried away the least of all. The main alkaline components in the dust from cement rotary kiln electrofilters are $K_2SO_4$ and $Na_2SO_4$. The high concentration of magnesium oxide predetermined preliminary experiments to establish the possibility of using dolomite–barium "tails" of the Khantagi tailings dump of the Achpolymetal JSC combined in silicate mixtures (1:1 and 1:10), which are shown in Figure 2, as well as changes in sample sizes.

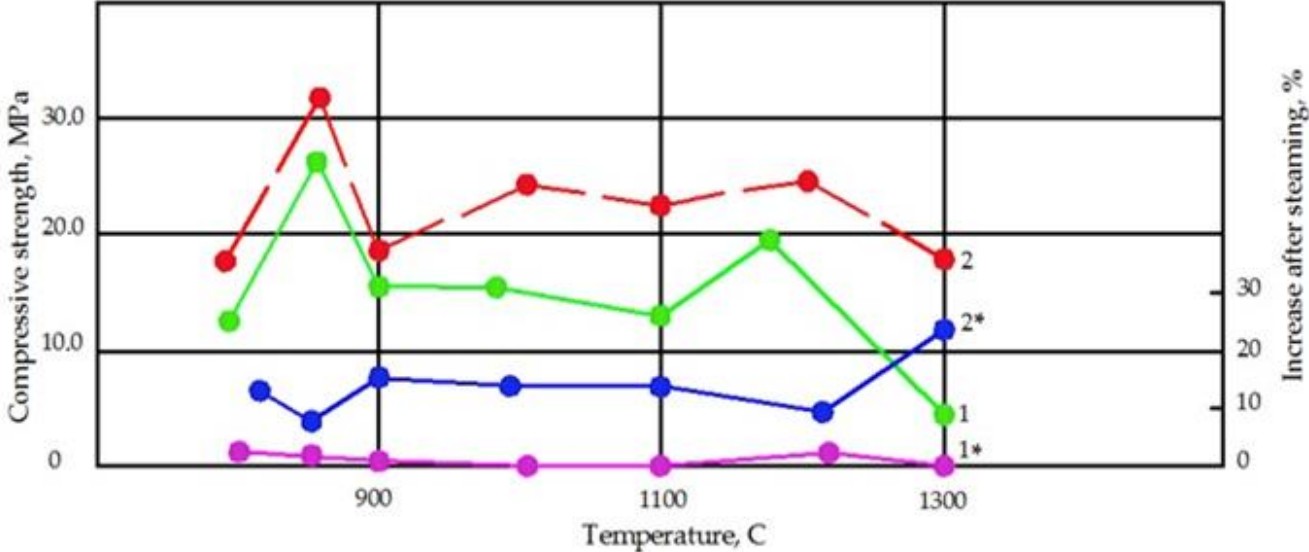

**Figure 2.** The effect of the binder firing temperature on the change in volume and strength: 1—the composition of the raw mixture 1:1; 2—the composition of the raw mixture 1:10; 1*—an increase in the volume of samples from the raw mixture 1:1 after steaming; 2*—an increase in the volume of samples from the raw mixture 1:10 after steaming.

The waste burned at different temperatures was ground to the same specific surface $S = 5000 \text{ cm}^2/\text{g}$. Then, the obtained binders were tested in autoclave hardening products. From these mixtures, samples were formed: cubes of $2 \times 2 \times 2$ cm at a pressing pressure of 20 MPa. The samples were subjected to autoclave treatment in an autoclave (Matest, Italy) according to the 3-6-3 h regime, and the pressure in the autoclave was 2 MPa. The moisture content of the charge was 15% of the weight of the dry mixture. Volsky sand, ground to a specific surface of $2500 \text{ cm}^2/\text{g}$, was used as a silica component. The results of measuring the volume of cubes and the data of the compression test of samples after steaming are shown in Figure 2.

The test results indicate that the behavior of the binder during the autoclave treatment depends on the temperature of its pre-firing and the ratio of binder:fill in the charge.

Deformation and cracking were not observed in samples containing 10% binder up to a firing temperature of 1300 °C

An increase in the size of the samples, the appearance of a grid of cracks and, ultimately, a decrease in strength occurred as a result of the hydration of the periclase during hydrothermal treatment.

An increase in the firing temperature from 900 to 1300 °C and an increase in the binder fraction up to 50% caused an increase in volume (up to 21%) and reduced the strength of the samples. Rankinite and mervinite bound calcium oxides into low-activity compounds, which were found in the waste firing products obtained at 1100 °C. Therefore, the use of such a binder causes a decrease in strength.

Subsequent firing to obtain an autoclave hardening binder was carried out at 800–900 °C. The silicate mass was prepared from waste burnt at a temperature of 800 °C: carbonate-barium tailings (8–12%), electrothermophosphoric slag (80–90%) or sand and dust from electric filters of Shymkent Cement JSC in an amount of 2–5% were used as an active hydraulic additive. Electrothermophosphoric slag was pre-crushed to a specific surface area of 300–500 m$^2$/kg.

The selected components of the raw mixture in their chemical and mineralogical composition can replace lime and sand in the traditional technology of silicate materials, and the dust from the electrofilters, which has a highly developed surface, can increase the ratio of the molding mixture, thereby creating conditions for the compensation of stresses arising in hardening products.

The components of the raw mixtures were mixed at a humidity of 8%. Beams 4 × 4 × 16 cm were formed from the prepared raw materials mixtures by vibrating (3 min), which were steamed in an autoclave according to the 3-6-2 mode at a pressure of 2.0 MPa. Earlier, we found that the vibrating method is most promising when forming a silicate mass with an increased content of magnesium oxide in the binder. The results of physical and mechanical tests of samples after steaming and after the autoclave treatment are given in Table 2. The products were obtained in factory conditions at the silicate brick factory (Shardara, Kazakhstan) according to the factory mode of the heat and humidity treatment.

**Table 2.** The composition of raw mixtures and the strength of products after steaming and autoclave treatment.

| Content of Components, wt. % | | Compressive Strength, MPa | |
|---|---|---|---|
| Electrothermophosphoric Slag | Burnt Dolomite-Barium Tails | After Steaming | After Autoclave Treatment at $p$ = 0.8 MPa According to the Mode 1.5–10–1.5 h |
| 85 | 15 | 20.5 | 43.1 |
| 70 | 30 | 24.0 | 53.9 |
| 65 | 35 | 21.6 | 47.1 |

The replacement of raw materials (lime and sand) in the production of silicate bricks with industrial waste made it possible to obtain products with a high strength of 45.2–49.3 MPa in contrast to the standard 41.9–43.0 MPa, as well as to expand the search for effective construction products and to reduce the cost of construction by using man-made waste.

As can be seen from Table 3, the use of a new binder made it possible to obtain very strong silicate samples (compositions 5, 6, 7): mixtures of polymetallic ore enrichment waste without additives of low strength.

**Table 3.** The composition of raw materials mixtures and the strength of the resulting products (silicate bricks) based on industrial waste.

| No. | Composition, wt. % | | | | | Compressive Strength after Autoclave Treatment, MPa |
|---|---|---|---|---|---|---|
| | Lime-Pushonka (for Comparison) | Burnt Dolomite-Barium Tails | Sand | Trapped Dust from Electric Filters | Electrothermo-Phosphoric Slag | |
| 1 | 12 | - | 88 | - | - | 26 |
| 2 | | 8 | 90 | 2 | - | 41.9 |
| 3 | | 10 | 86 | 4 | - | 43 |
| 4 | | 12 | 83 | 5 | - | 40.8 |
| 5 | | 8 | - | 2 | 90 | 46 |
| 6 | | 10 | - | 4 | 86 | 49.3 |
| 7 | | 12 | - | 5 | 83 | 45.2 |
| 8 | | 100 | - | - | - | 1.9 |

According to an X-ray phase analysis of autoclave materials, the cementing substance in silicate products based on the binder (composition 1) consists of calcium hydrosilicates of type CSH(B) d = 0.304 nm and tobermorite d = 0.307 nm. An X-ray phase analysis of hydration products in waste-based samples (8) shows that the main hydration products are $Ca(OH)_2$, $Mg(OH)_2$, tobermorite and CSH(B), and a diffraction maximum of d = 0.210 nm belonging to $BaSO_4$ is noted.

When ground sand and dust from electrofilters (composition 3) are added, the diffraction maxima belonging to $Ca(OH)_2$ completely disappear and reflections characteristic of CSH(B) appear, and the concentration of this hydrosilicate is higher than in composition 8. The reflections belonging to $Mg(OH)_2$ are small, and apparently part of $Mg(OH)_2$ binds to a hydrosilicate of the serpentine composition d = 0.369; 0.247; 0.457; 0.152 nm. Thus, when lime is replaced with "waste", the interaction occurs not only between CaO and $SiO_2$, but also between MgO and $SiO_2$ with the formation of serpentine, which leads to an increase in the strength of products.

The phase composition of the cementing agent in the sample (composition 6) based on a binder containing burnt tails, electrothermophosphoric slag and trapped dust from electrofilters is characterized by a complex of low-base calcium hydrosilicates. An X–ray phase analysis shows the presence of CSH(B)—d = 0.302; $C_5S_5H$—d = 0.275; 0.311; 0.365; 0.205 nm; and tobermorite—d = 0.307 nm.

The simultaneous presence of fibrous and crystalline calcium hydrosilicates of various compositions in hydration products is the main factor that leads to the production of products with maximum strength [15–25].

Calcium hydrosilicates of fibrous (filamentous) shape give products an increased tensile and bending strength, impact strength and crack resistance. They perform the role of microarmature and prevent the spreading of cracks [15–25].

The electronic micrograph (Figures 3 and 4) shows hydration products in the form of felt-like weaves, which ensures a high strength of the samples. Products obtained on the basis of binder 3 and 6 (Table 3) have a higher strength of 19.2–23.3 MPa than products obtained on the basis of lime.

The role of alkaline cations in the formation of fibrous forms of calcium hydrosilicates during the autoclave hardening of a lime–silica binder is that, in the presence of small alkali additives, the concentration of silica in the liquid phase increases and the concentration of $Ca(OH)_2$ decreases. Thus, conditions are created for the existence of associated silicon–oxygen complexes that contribute to the formation of calcium hydrosilicates with highly polymerized chain and ribbon silicon–oxygen anions, where the filamentous shape of their crystals are due to the peculiarities of their crystal chemical structure [26].

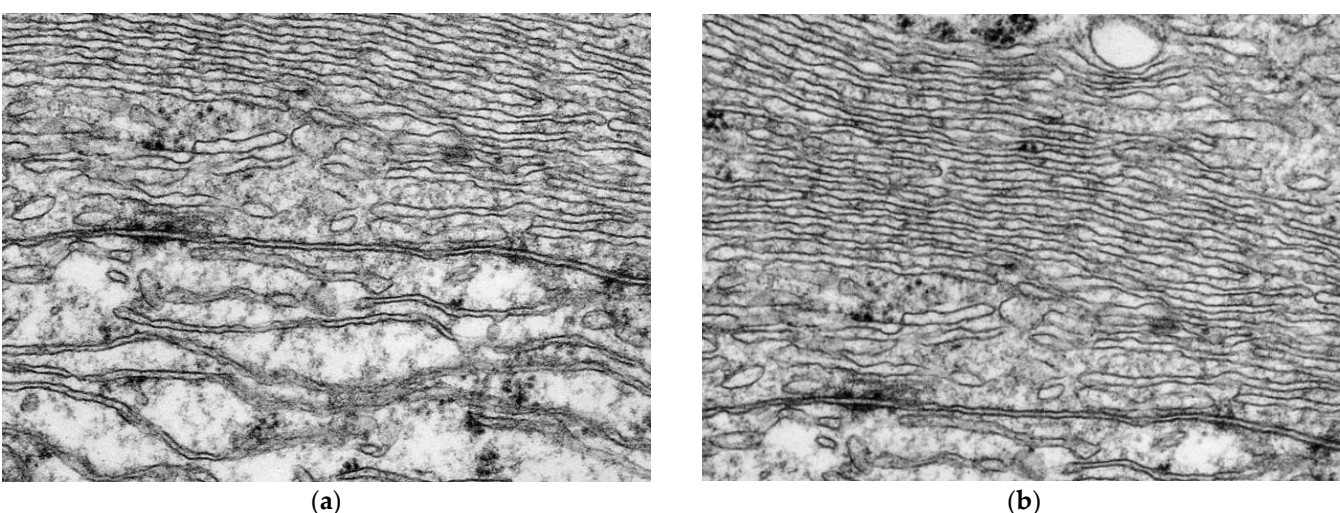

(**a**)                        (**b**)

**Figure 3.** Electronic micrographs (magnification × 200 times) from the chips of products after autoclave treatment obtained from the masses: (**a**) composition 8; (**b**) composition 1.

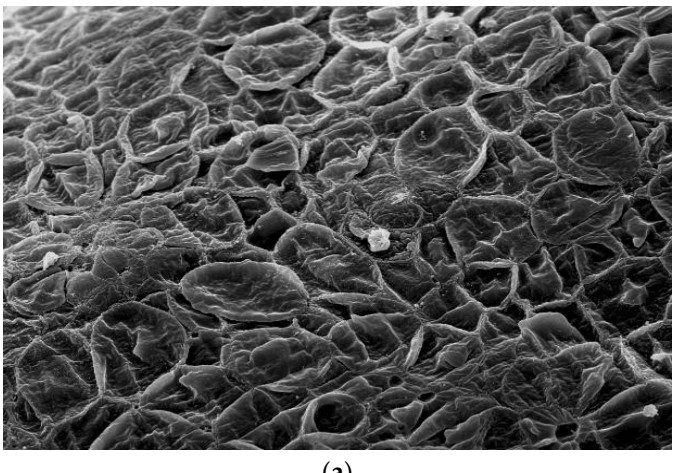
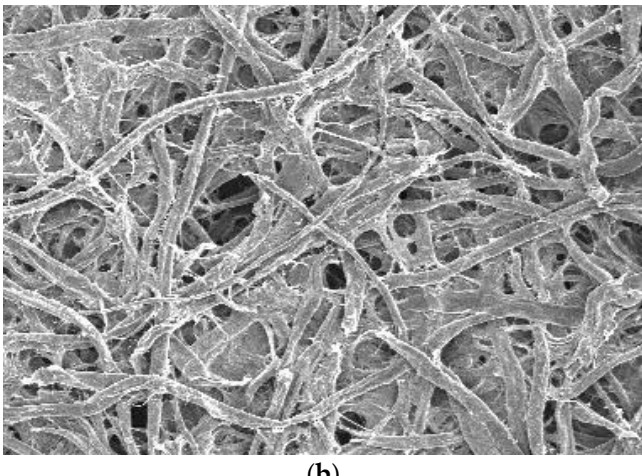

(**a**)                                                                                    (**b**)

**Figure 4.** Electronic micrographs (magnification × 2500 times) from the chips of products after autoclave treatment obtained from the masses: (**a**) composition 3; (**b**) composition 6.

The presence of alkaline cations in the dust from the electrofilters used in the composition of the proposed raw material mixture contributes to the creation of conditions for the existence of associated silicon–oxygen complexes that contribute to the formation of calcium hydrosilicates with highly polymerized chain and ribbon silicon–oxygen anions.

The developed binder compositions were tested in silicate products for frost resistance and sulfate resistance. The results of the durability determinations are given in Table 4.

**Table 4.** Characteristics of composite products obtained on the basis of industrial waste.

| Train No. | Limit of Samples under Compression, MPa | | | Softening Factor | Frost Resistance Coefficient |
|---|---|---|---|---|---|
| | After Autoclaving, MPa | 12 Months of Hardening | | 200 Cycles of Freezing and Thawing | |
| 1 | 26 | 28.4 | 25.4 | 20.32 | 0.9 | 0.8 |
| 3 | 43 | 45.3 | 42.1 | 35.7 | 0.93 | 0.85 |
| 5 | 46 | 47.5 | 45.0 | 39.15 | 0.94 | 0.87 |
| 6 | 49.6 | 50.8 | 47.9 | 41.7 | 0.94 | 0.87 |

Freezing was carried out in the climate chamber SM-70/180-2000 (Moscow, Russia) at −17–20 °C for four hours, and then the samples were thawed at 17 + 3 °C for three hours.

The values of frost resistance and softening coefficients confirm the high quality of silicate concrete products when using a new binder. According to GOST 379-95, the brand of brick corresponds to the compressive strength: 30; 25; 20; 15; 12,5; 10; and 7.5 MPa. The latter value is only for hollow bricks. The brick must hold at least 15 cycles of freezing and thawing.

The proposed formulations are characterized by a high sulfate resistance. The samples were tested in three aggressive solutions: sodium sulfate (3%), magnesium sulfate (3%) and magnesium chloride (5%) for twelve months.

At the same time, the bending strength of these samples decreased only by 4–5% under the influence of sodium sulfate, and, in solutions of magnesia salts, by 16–18%. The increased resistance of magnesian silicate masses is due to the appearance of magnesium hydrosilicates, which are not subjected to magnesian aggression. Samples of lime-based silicate concrete proved to be minimally resistant to the action of aggressive magnesia solutions: after 12 months, their strength decreased by 50–55%.

Among the construction wall materials produced in Kazakhstan, silicate brick currently occupies one of the leading places and has an undoubted prospect for the future

because its production, unlike the production of clay bricks, has significant technical and economic advantages, such as the simplicity of the manufacturing process, a high level of mechanization and lower fuel consumption. In addition, the cost and labor intensity of its manufacturing is two times lower, and the duration of its production cycle is 5–10 times less. The quality of silicate bricks is higher, and the dimensions and appearance are always maintained in accordance with GOST.

The economic and technological feasibility of using the low-temperature roasting of technogenic waste instead of lime as a binder for autoclave products lies in the fact that it is not required to produce the mining and blasting of rocks, and there are no crushing stages. The process uses a low firing temperature (700–900 °C), and the firing products do not require additional grinding because there is a self-dispersion of the material during firing at 700–900 °C. which all give a significant economic effect with a simultaneous environmental effect due to the associated utilization of man-made industrial waste, making this technology resource-saving.

These studies correlate with previous studies [15–30] and bring their novelty and practical significance to the field of composite building materials.

## 4. Conclusions

Based on the analysis of experimental data obtained by using high-magnesian lime in the production of autoclave hardening products, the following conclusions were made:

- When using steam in the autoclaving process and the plastic molding method, it is possible to obtain products of sufficient strength from lime–sand mixtures based on magnesia and dolomite lime;
- It was found that MgO acquires the greatest hydration ability after firing at a temperature of 800 °C;
- The composition of a composite silicate mass based on burnt carbonate-barium tailings (8–12%), electrothermophosphoric slags (82–90%) or sand and captured dust from electrofilters of cement plants (2–5%) used to obtain composite products of autoclave hardening was developed;
- It was found that the cementing substance in composite silicate products is represented by CSH(B) hydrosilicate: tobermorite and serpentine. The simultaneous presence of fibrous and crystalline calcium and magnesium hydrosilicates in hydration products leads to the production of composite products with a maximum strength of Rsf = 41–49 MPa;
- A composite product based on a new binder from man-made waste is characterized by a higher strength and frost and sulfate resistance in contrast to autoclave materials based on lime;
- Sand can be successfully replaced with electrothermophosphoric slag, and, as an active hydraulic additive, captured dust from the electrofilters of cement plants can be used;
- The presence of alkaline cations in the dust from the electrofilters used as part of the proposed raw material mixture contributes to the creation of conditions for the existence of associated silicon–oxygen complexes that promote the formation of calcium hydrosilicates with highly polymerized chain and ribbon silicon–oxygen anions that increase the strength characteristics of the developed composite product by 19.2–23.3 MPa.

**Author Contributions:** Writing—review and editing, S.A., U.A. and O.K.; methodology, B.S. and G.S.; formal analysis, N.S. and E.A.; investigation, G.K.; supervision, A.A. and I.P.; data curation, Z.A.; resources, T.A.; validation, S.U. All authors have read and agreed to the published version of the manuscript.

**Funding:** This research received no external funding.

**Data Availability Statement:** Not applicable.

**Conflicts of Interest:** The authors declare no conflict of interest.

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
