# Peer review of "Man-Made Raw Materials for the Production of Composite Silicate Materials Using Energy-Saving Technology"

_jcs, doi:10.3390/jcs7030124_

Round 1

Reviewer 1 Report

"The article "Man-made raw materials for the production of composite silicate materials using energy-saving technology" discusses the synthesis of composite silicate materials from man-made waste. While the motivation for the research is intriguing, the experiments and results do not meet the publication standards. I cannot recommend the publication of this article in the Journal at this time.

To improve the manuscript for publication, I suggest the authors revise it with the following considerations:

  1. Improve the presentation of the results.

  2. Expand the results section.

  3. Improve the figure quality, particularly Fig. 1, to ensure it is properly visible.

  4. Correct any errors in the references.

  5. Revise the conclusion to make it more comprehensive.

I recommend the authors extensively revise the manuscript before resubmitting it."

Author Response

Good afternoon, Dear Reviewer! 

We have reworked our paper, improved the presentation of results, expanded and improved sections, corrected errors and revised the conclusion. The corrected version has been uploaded.

1. Improved the presentation of the results.
2. The section has been expanded.
3. In our opinion, the quality of the drawing is now 2 sufficient. It will not be possible to improve it, unfortunately it is not available to us today, since the colleague responsible for the experimental data (the source data and drawings in his laptop) has been in the hospital for the second week after a serious accident). Please be understanding.
4. Corrected errors and checked the text from a native English speaker.
5. The conclusion has been revised and improved.

Thank you again for your valuable comments and suggestions.

Reviewer 2 Report

English need to be improved (average 85%). 

A major improvement needs to be made.

Other details are described in the attached word file.

Author Response

Good afternoon, Dear Reviewer! 

We have reworked our paper, improved the presentation of results, expanded and improved sections, corrected errors and revised the conclusion. The corrected version has been uploaded.

Detailed answers are described in the attached file below.

Thank you again for your valuable comments and suggestions.

Reviewer 3 Report

The manuscript presents the development of composite silicate mass compositions based on man-made waste for the production of autoclave hardening products, as well as the results of physico-chemical studies of hydration products of the silicate materials.

The topic of waste reuse is of significant environmental impact.

However, the manuscript was not well-prepared: there lacks detailed experimental section; oddly in Results section the authors introduced the experiments, which should be introduced in Section 2. Which instruments were used is not clear, as there is no product information (serial No., brand, company, city, country). Hence the overall reproducibility is low. English usage needs significant improvement.

In conclusion, a major revision is needed before further consideration in JCS.

More details for consideration:

1. Introduction: first paragraph needs to be re-written because it does not read well.

2. There lacks introduction of the objectives of the study. If these principals in L67 are for general works in the industry, there should be references.

3. L96, what does “proper composition” refer to?

4. English terms are vague, please check carefully, for example “highly magnesian”, “…are located more evenly…” etc.

5. “Generally accepted methods and methods of laboratory testing of silicate” is too vague, please specify which methods were used with details.

6. The title states “energy-saving technology” which was not elucidated in the manuscript.

7. Please cite doi: 10.1016/j.jnoncrysol.2016.06.027 as a reliable reference for the crystalline and amorphous structures of the calcium silicates and cement-forming properties.

Author Response

Good afternoon, Dear Reviewer! 

A detailed experimental section has been added, serial number, brand, company, city, country of the equipment used have been added to improve their reproducibility. English is improved by a native speaker.

  1. The introduction has been edited and reviewed.
  2. The purpose of the study was added.
  3. Corrected.
  4. Edited and changed
  5. Changed and brought, specifying methods.
  6. At the end of section 3, in the penultimate paragraph, the energy-saving effect of the developed technology is indicated.
  7. The source is given.

Thank you again for your valuable comments and suggestions.

Round 2

Reviewer 1 Report

The manuscript can be accepted now.

Author Response

Good afternoon, Dear Reviewer!

Thank you again for your valuable comments and suggestions . We wish you health and creative success!

Reviewer 2 Report

Some corrections have been made, and still, some parts are to be improved.

Author Response

Good afternoon, Dear Reviewer!

We have accepted all the corrections and they are fixed in the article with highlights.

The answers to the comments are attached below.

Thank you again for your valuable comments and suggestions . We wish you health and creative success!

Reviewer 3 Report

It appears all issues have been addressed. However, the integrity is affected by the "tracked changes". Please "accept all changes" and submit a clean version, with changes highlighted.

Author Response

Good afternoon, Dear Reviewer!

We have accepted all the corrections and they are fixed in the PDF version of the article with highlights. We also reflected all the changes in color in the article.

Thank you again for your valuable comments and suggestions . We wish you health and creative success!

Round 3

Reviewer 3 Report

It can be published.